

# Coupling ocean currents and waves for seamless cross-scale modeling during Medicane Ianos

Salvatore Causio[1], Seimur Shirinov[1,2], Ivan Federico[1], Giovanni De Cillis[1], Emanuela Clementi[1], Lorenzo Mentaschi[2], Giovanni Coppini[1]

[1]CMCC Foundation - Euro-Mediterranean Center on Climate Change (CMCC), Italy, Lecce
[2]University of Bologna, Italy, Bologna

*Correspondence to*: Salvatore Causio (salvatore.causio@cmcc.it)

**Abstract.** This study investigates the effects of a two-way wave-circulation coupled modeling framework during extreme weather events, with a particular focus on Medicane Ianos, one of the most intense cyclones occurred in the Mediterranean Sea. By utilizing a high-resolution unstructured numerical grid, the study explores wave-
current interactions in both open ocean and coastal environments. To this scope, we developed the first external coupler dealing with the SHYFEM-MPI circulation model and the WAVEWATCH III wave model. The interactions considered in this framework include sea-state dependent momentum flux, radiation stress, Doppler shift, dynamic water depth for waves, and effective wind speed. The study adopts a rigorous validation of the formulations using idealized benchmarks tailored for these specific processes. Afterwards, the modeling
framework was employed in real-case simulations of Medicane Ianos. The model is calibrated, and the ocean variables are rigorously validated against in-situ and Earth Observation (EO) data, including satellite-based measurement. The study found that wave-induced surge components contribute from 10 to 30% of the total water level during the storm, and that sea-state dependent momentum during a Medicane can influence the vertical structure of the ocean up to 100 m. The accuracy of the wave model improves by around 3% in terms of RMSE
when coupled with a circulation model.

This study underscores the importance of such coupled models in accurately forecasting Medicanes, storm surges, and their impacts, particularly as climate change intensifies extreme events in the Mediterranean Sea.

### 1. Introduction

Advanced knowledge on surface wind-driven gravity waves and currents, and their interactions in the ocean is of great relevance to many decision support systems, such as meteocean forecasting, early warning for storm surge events, oil spill evolution, search and rescue, beach erosion, site choices for offshore infrastructures (Hashemi and Neill, 2014). Physically, the surface gravity waves can affect the vertical mixing, as well as surface and bottom stresses experienced by currents. Surface waves and currents can also exchange energy through the concept of
radiation stress (Longuet-Higgins and Stewart, 1961; Mellor, 2003) or vortex force (VF) (McWilliams et al., 2004; Ardhuin et al., 2008). In return, ocean currents can modify the relative speed of the air above the sea surface (relative wind effect) and change the absolute frequency of waves due to the Doppler shift. Spatial variability of currents can modify the relative wave frequency and cause wave refraction, shoaling, and breaking that mimic bathymetric effects.


An accurate computation of the Sea Level (TWL) during storm surge events combining multiple drivers (e.g. surges, tides, waves) becomes relevant to forecast flooding and impacts in coastal and urban areas. Kim et al. (2008) used a coupled surge-wave-tide (SuWAT) model to demonstrate that high tides can mitigate storm surges. Their study highlighted the relationship between locally generated surge and wave setup with the tidal state. They
emphasized that considering surges, waves, and tides independently can lead to an overestimation of TWL Additionally, they underscored the importance of incorporating bottom friction, wind-wave interactions, and wave-current interactions for more accurate storm surge estimations. Continuing the development, with the focus on the effects of wind stress parameterizations and wave-induced radiation stress on storm surge, Kim et al. (2010) estimated 40% contribution of radiation stresses to the peak sea level rise.



Jo et al. (2024) studied the typhoon-caused inundations concluding that uncoupled models overestimate wave-overtopping peak discharge by 8%. Hsiao et al. (2019) used fully coupled circulation-wave model (SCHISM + WW3) to simulate super typhoon events. They observed that surface waves, especially wave setups exhibited the most significant nonlinear interaction with storm tides, while wave-tide interaction was relatively minor. Dietrich et al. (2011) coupled a shallow-water model (ADCIRC) with the wave model SWAN, utilizing the same
unstructured grid to simulate the waves and storm surges induced by the hurricanes Katrina and Rita in the Gulf of Mexico. The model demonstrated that as the hurricane Katrina moved onto the continental shelf, depth-limited breaking waves exerted stresses on the water column, increasing overall water levels by 0.2–0.3 m and driving currents, contributing up to 35% of the TWL and 10% of overall currents in the region.

Extreme meteorological events are intensifying in the Mediterranean Sea due to climate changes, among these we have the Mediterranean tropical-like cyclones or Medicanes. (Cavicchia et al., 2014; Sánchez-Arcilla et al., 2016). Medicanes are rare meteorological phenomena that exhibit characteristics similar to tropical cyclones, such as a warm core, eye formation, and strong winds. These systems, however, form and evolve under the unique conditions of the Mediterranean Sea, typically during the autumn and early winter months when sea surface
temperatures are sufficiently warm to support their development (Cavicchia et al., 2014). Although Medicanes are generally less intense and shorter-lived compared to their tropical counterparts, they can still cause significant damage and pose considerable risks to the regions they affect. Notable examples include Medicane Rolf in 2011 and Medicane Zorbas in 2018, both of which provided valuable insights into the dynamics and impacts of these systems (Flaounas et al., 2022). Among the most harmful recent Medicanes is Medicane Ianos, which struck
Greece in September 2020, causing widespread damages, exceeding 700 millions of dollars and resulting in human casualties (Diakakis et al., 2023). The study of Medicanes is crucial for advancing our understanding of these phenomena and enhancing preparedness and response strategies in the Mediterranean region.

This study aims to investigate the impact of a two-way coupled wave-circulation model on ocean dynamics during extreme events, with a focus on Medicane Ianos. Using very high-resolution and unstructured numerical grid, we describe the contribution of wave-currents interaction during one of the most intense cyclones in the Mediterranean Sea.
We carried out this work using a new modeling framework based on SHYFEM-MPI (Micaletto et al., 2022; Verri et al., 2023) and a state-of-the-art community wave model WAVEWATCH III (hereafter WW3) (Tolman, 2009;
WW3DG, 2019). For this study, an external *Python*-based coupler is developed to manage the mutual exchanges between the numerical cores, enabling two-way coupling.
The WW3 model is widely used by several organizations worldwide to simulate waves in various systems across different regions, ranging from global (Mentaschi et al., 2020; Alday et al., 2021; Brus et al., 2021) to regional (Abdolali et al., 2020; Causio et al., 2021, 2024) and coastal scales (Pillai et al., 2022a, 2022b).

The SHYFEM model has previously been coupled with wave models, as in Roland et al. (2009). It serves as the circulation core of the storm surge early warning system Kassandra (Ferrarin et al., 2013) and has recently been employed to analyze uncertainties related to the simulation of the Medicane Ianos (Ferrarin et al., 2023). In these studies, they adopted a barotropic implementation, with the unstructured WWMIII wave model (Roland, 2009) integrated through hard-coupling.

Present work incorporates new processes into the two-way coupling SHYFEM-WW3, including the sea state dependent momentum flux and wind field correction based on the air stability parameter. This marks the first study to implement a three-dimensional baroclinic version of wave coupled SHYFEM model.

The reliability of the results and the code are validated using well-established idealized benchmarks and analytical results. Specifically, we use the planar beach (Xia et al., 2020) and the wave-tidal-driven inlet (Cobb and Blain,
95   2002).
Furthermore, this newly developed coupled system is applied to a real-case, simulating the impacts of Medicane Ianos on ocean and wave fields, as well as their interactions, using a cross-scale unstructured-grid domain. This domain enables us to examine both (i) open ocean features, such as sea surface cooling induced by the hurricane,



the impact of the coupling on vertical profiles, and the effects of circulation on large scale wave dynamics, and
(ii) local coastal features, including TWL and wave contribution to storm surge.

The paper is structured as follows. Section 2 introduces the modelling framework, including the circulation and
wave modelling suite as well as the coupling methodology. Section 3 validates the coupling procedure using
idealized test cases. Section 4 investigates the Medicane induced effects on the ocean, with a specific focus on
coastal storm surge. Finally, the results are presented in Section 5, followed by the conclusions in Section 6.

## 2.  The modelling framework

This study is based on a coupled modeling framework built on WW3 and SHYFEM-MPI. Both models run on the
same unstructured computational grid, a classic methodology in coupled unstructured modeling, used from early
studied to the most recent ones, such as in Mentaschi et al. (2023). This avoids further interpolations, and it allows
a seamless approach to address large-scale and coastal processes. Unstructured grids also ensure an accurate
representation of both natural and artificial irregular coastal features (Bertin et al., 2014), and it allows for high-
resolution modelling along the coast.

### 2.1.  Circulation modelling

The circulation model used in this study is SHYFEM (System of HydrodYnamic Finite Element Module), a 3D
baroclinic finite-element hydrodynamic model (Umgiesser et al., 2004; Federico et al., 2017; Verri et al., 2023)
that solves the Navier–Stokes equations by applying the hydrostatic and Boussinesq approximations. The
SHYFEM-MPI version, developed by the Centro Euro-Mediterraneo sui Cambiamenti Climatici (CMCC)
Micaletto et al. (2022), is used in this study and, for the first time, is coupled with the wave model WW3.
The model employs a semi-implicit algorithm for time integration, providing unconditional stability with respect
to gravity waves, bottom friction, and Coriolis terms, thus allowing transport variables to be solved explicitly.
The Coriolis term and pressure gradient in the momentum equation, along with the divergence terms in the
continuity equation, are treated semi-implicitly. Bottom friction and vertical eddy viscosity are treated fully
implicitly for stability reasons, while the remaining terms (advective and horizontal diffusion terms in the
momentum equation) are treated explicitly. The finite element discretization is applied on unstructured B-type
grids. More detailed descriptions of the model equations and discretization methods are provided in Micaletto et
al. (2022) and Verri et al. (2023).
In this work, we implemented the transport and diffusion equation for tracers, with horizontal and vertical
advection values based on a total variation diminishing (TVD) scheme. An upwind scheme is employed to
discretize the horizontal advection of momentum, and the Smagorinsky (1963) formulation is used to compute
horizontal eddy viscosity. Vertical viscosities and diffusivities are calculated using a k-ε scheme adapted from the
General Ocean Turbulence Model (GOTM) (Burchard and Petersen, 1999). Bottom stress is computed using the
quadratic formulation with a bottom drag coefficient defined based on the logarithmic formulation and water
depth. In the logarithmic formulation, the von Karman constant is assigned a value of 0.4, and the bottom
roughness length is 0.01.
The model has been successfully applied in operational (Federico et al., 2017) and relocatable forecasting systems
(Trotta et al., 2021), storm surge events (Park et al., 2022; Alessandri et al., 2023), and as a part of a digital twin
by (Pillai et al., 2022b).


### 2.2.  Wave modelling

The wave model used is WW3 v6.07, a third-generation wave modeling framework that incorporates the latest
scientific advancements in wind-wave modeling and dynamics. Developed by the US National Centers for
Environmental Prediction (NOAA/NCEP) and inspired by the WAM model (Komen et al., 1984), WW3 solves
the random phase spectral action density balance equation for wavenumber-direction spectra. The model includes
options for shallow-water (surf zone) applications and features numerical schemes capable of solving the
propagation of a wave spectrum at high resolutions on unstructured (triangular) grids (Abdolali et al., 2020).
In this study, the wave energy spectra are discretized into 24 equally spaced directional bins covering the full
circle, and a 30 log-spatially varying wavenumber range (0.0500–0.7932 Hz). The source input and dissipation





terms are based on the Source Term 4 (ST4) physics from Ardhuin et al. (2010). Non-linear wave-wave
       interactions are implemented using the Discrete Interaction Approximation (DIA) from (Hasselmann and
       Hasselmann, 1985). Bottom friction is defined using the JONSWAP parametrization (Hasselmann et al., 1973),
       and depth-induced breaking follows the criterion described by Battjes and Janssen (1978) and Miche (1944) for
       defining the breaking threshold. Furthermore, we calibrated the wind-wave coupling parameter (BETAMAX) to
improve model results and reduce the underestimation of wave height. Nonlinear triad interactions are modeled
       using the Lumped Triad Approximation (LTA) model of Eldeberky (1996). Second-order propagation scheme
       Contour Residual Distribution – Positive Streamline Invariant (CRD-PSI) (Roland, 2009) is used to describe wave
       propagation with a global timestep of 200 s.

**2.3. Coupling methodology**
       The two-way interaction in the modeling framework is achieved via an external *Python*-based coupler, which has
       the advantage of using the native versions of the standalone models, thereby minimizing code changes. The
       primary changes applied to the native codes relates to new physical processes inclusion.
       For this study, multiple sensitivity tests are conducted to evaluate the system's response to the coupling frequency
($\Delta t_{coupling}$), identifying 1 hour as an optimal balance between the exchange frequency and the computational
       performance. However, $\Delta t_{coupling}$ is set to 30 minutes for the idealized test cases to be consistent with previous
       studies, while 1 hour is used for the real case.
       The main objective of this study is to characterize the wave-induced surge component in the total water level,
       with a focus on the effect of waves on circulation. This involves incorporating the theory of Longuet-Higgins and
Stewart (1961). Additionally, to provide a more comprehensive set of interactions from coastal to large scale, we
       include sea-state-dependent momentum flux, based on the neutral drag coefficient, as described by Clementi et
       al. (2017). Conversely, to preserve the integrity of two-way interactions, the effect of circulation on waves is
       addressed by incorporating the Doppler shift induced by surface ocean currents, by dynamically updating the
       water depth based on the sea-surface height and by correcting the wind field according to a stability parameter
dependent on the air-sea temperature difference (Abdalla and Bidlot, 2002). This approach accounts for wind
       gustiness, which can enhance the representation of extreme conditions during storms (Abdalla and Cavaleri,
       2002).
       A schematic representation of the two-way coupling is shown in Figure 1 and detailed below.
       The circulation model receives the radiation stress components and the wind neutral drag coefficient computed
by the wave model. The radiation stress components *Sxx*, *Sxy* and *Syy* account for wave transformation in coastal
       areas, resulting in a net flux of momentum. According to the Longuet-Higgins and Stewart (1962) theory, wave
       setup balances the shoreward gradient in radiation stress, with the sea surface pressure gradient counteracting the
       gradient of incoming momentum. The wave-induced momentum flux (*F*), derived by the gradients of the radiation
       stress is estimated and included in the momentum equation (Eq. 1).

$$F = \frac{1}{\delta}\left(\frac{\partial S_{xx}}{\partial x} + \frac{\partial S_{xy}}{\partial y}, \frac{\partial S_{xy}}{\partial x} + \frac{\partial S_{yy}}{\partial y}\right) \qquad \text{1)}$$

The sea-state dependent wind stress involves exchanging the wind-neutral drag coefficient from the WW3 wave
       model, following Clementi et al. (2017). This coefficient is computed using a formulation derived from the quasi-
       linear theory of wind-wave generation, as developed by Janssen (1989, 1991) and based on Miles (1957).
       According to this theory, the neutral drag coefficient ($C_{Dn}$) (Eq. 2) depends on the effective roughness length,
       which is influenced by the sea state through the wave-induced stress estimated from the wave spectra.

$$C_{Dn} = \left(\frac{\kappa}{log(z_u/z_0)}\right)^2 \qquad \text{2)}$$

Where $z_u$ is the height of the wind forcing, $\kappa = 0.4$ is the von Karman constant, and $z_0$ is the roughness length
       modified by the wave-supported stress $\tau_w$ as in Eq. 3:

$$z_0 = \frac{\alpha_0 \tau}{\sqrt{1 - \tau_w/\tau}} \qquad \text{3)}$$



Where $\alpha_0 = 0.01$. The $CDn$ is transferred to the circulation model for the estimation of the turbulent drag coefficient according to Large (2006).

The circulation interacts with the wave field through sea level, currents, and air-sea temperature difference. The sea level ($\eta$) allows for a dynamic adjustment ($d$) of the water depth ($h$) in the wave equations as in Eq. 4:

$$d = h + \eta \qquad\qquad 4)$$

Ocean currents influence wave characteristics, including wavelength, amplitude, wavenumber, and direction. In the WW3 model, the Doppler shift accounts for the changes in wave frequency due to the relative motion between waves and currents, which can be expressed as in Eq. 5:

$$\omega = \sigma + \mathbf{k} \cdot \mathbf{U} \qquad\qquad 5)$$

$\omega$: absolute frequency, $\sigma$: relative frequency, $\mathbf{k}$ : vector of wave number, $\mathbf{U}$: current velocity.

The stability of the atmosphere is influenced by the air-sea temperature difference ($\varDelta T$). Abdalla and Bidlot (2002) formulated a stability correction replacing the wind speed in WW3 with an effective wind speed, ensuring that the wave growth correspond to both stable and unstable wave growth patterns. The $\varDelta T$ defines a stability parameter
according to Eq. 6:

$$ST = \frac{hg}{Uh^2}\frac{\varDelta T}{T_0} \qquad\qquad 6)$$

$Uh$: wind speed at height $h$, $\varDelta T$: air-sea temperature difference, $T_0$: reference temperature. $ST$ is used to compute effective wind speed, $Ue$ (Eq. 7):

$$U_e = U_{10}\left(\frac{c_0}{1 \pm c_1 tanh[c_2(ST - ST_0)]}\right)^{\frac{1}{2}} \qquad\qquad 7)$$

$U10$: wind speed at 10m, $c_0$: 1.4, $c_1$: 0.1, $c_2$: 150, and $ST0$: -0.0.1

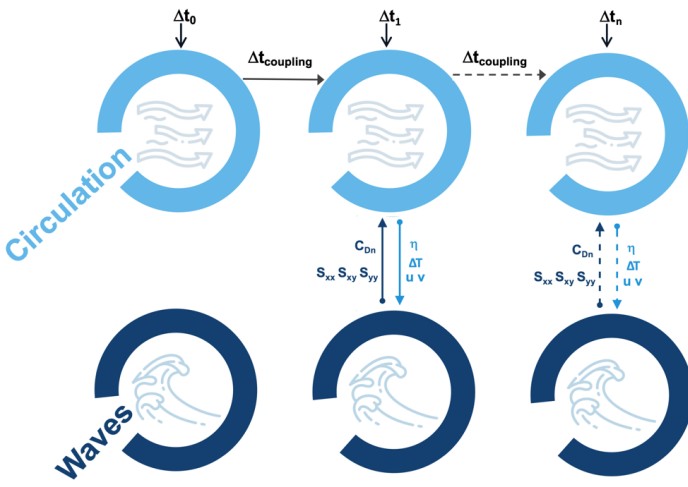


**Figure 1 The coupling procedure between SHYFEM-MPI and WW3. When the simulation starts, models exchange information after the first time step (e.g. $\Delta t_0$=300s), then exchange every $\Delta t_{coupling}$ (e.g. every 1h). SHYFEM-MPI sends sea level (h), current fields (u,v) and air-sea temperature difference ($\Delta T$) to WW3. WW3 sends neutral drag coefficient**
**($C_{Dn}$) and radiation stresses ($S_{xx}, S_{xy}, S_{yy}$) to SHYFEM-MPI.**





### 3. Validation of the coupling strategy with idealized testcases


The robustness of the two-way coupling and its physics is assessed and evaluated through idealized numerical experiments. The test cases selected for this assessment include two common benchmarks for the wave-currents interaction at coastal scale: a planar beach (Figure 2(a)) from (Xia et al., 2020) and an idealized tidal inlet (Figure 2(b)) from (Cobb and Blain, 2002).

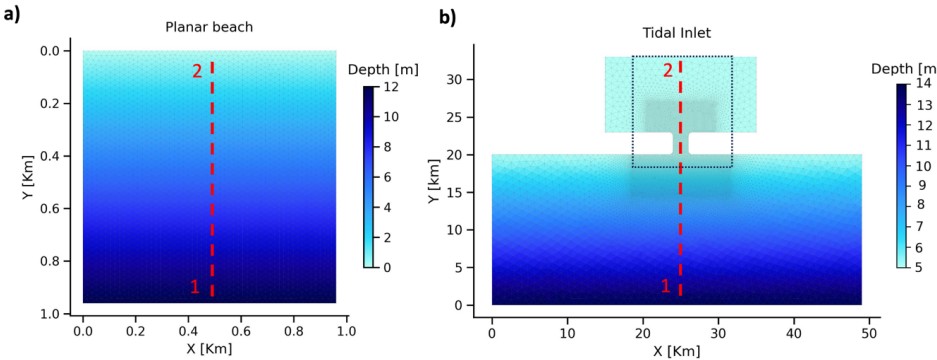


**Figure 2 Numerical domain and bathymetry of the two idealized benchmarks: the planar beach (a) and the tidal inlet (b). Dashed red lines show the transect from point 1 to point 2 where the numerical solutions were assessed. The dotted box in b) denotes the area magnified and shown later in the "Results" section.**

Table 1 lists the acronyms of the simulations used for the validation of the idealized testcases.


**Table 1 Configurations and nomenclature for the numerical setup used in this study.**

| Acronym | Description |
| --- | --- |
| *W-0* | Standalone wave model |
| *C-0* | Standalone circulation model |
| *W-C* | two-way coupled system Wave-Circulation |
| *C-WF* | Circulation model forced by Wave |

### 3.1. The planar beach

The planar beach is used in this work for assessing the wave setup/setdown along the transect 1-2 (Figure 2(a)). The results obtained from our numerical model are compared with the analytical solution as described in Xia et al. (2004), to validate the accuracy and reliability of the implementation in modeling wave contributions to the surge, ensuring that the modeled outcomes are consistent with the theoretical solution. The validation is carried out utilizing the C-WF configuration, where the wave model exclusively provides inputs (forcing fields) to the

circulation model. This setup allows the model to reach a semi-stationary condition more quickly.

The planar beach is a gently sloping square basin with a side length of 960 m, as illustrated in Figure 2(a). The bathymetry is constant along the shore and slopes linearly from 0 to 12 m in the cross-shore direction. The numerical grid has 5400 elements and 2797 nodes, and a horizontal resolution of approximately 20 m. The vertical discretization consists of 12 z-levels, each with a thickness of 1 m. The wave model is forced along the open

boundary by waves with a 2 m significant wave height (SWH), a peak period of 10 seconds, and a 10-degrees oblique angle relative to the shore-normal direction. The circulation model starts from a resting state and is only



influenced by the radiation stress from the WW3 model, with no other external forcings applied. These simulation conditions derives from Xia et al. (2020).

As the waves propagated from the offshore open boundary within the model domain, they start to shoal and breaking approximately 600 m from the shore, at which point the depth-induced breaking criterion is met. As shown in Figure 3, the simulated surface elevation produced by the circulation model match properly with the analytical solution (Xia et al., 2004; Wang and Sheng, 2016).

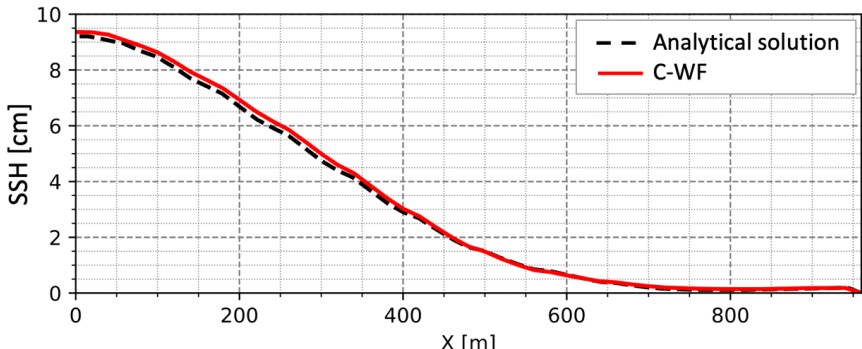

**Figure 3 Planar beach testcase: comparison between model and analytical solution along the transect 1-2 shown in**
**Figure 2(a). The testcase verifies the C-WF simulation, assessing the role of wave set-up (via radiation stresses) to the total water level.**

### 3.2 The coastal inlet

A second benchmark defined as coastal inlet, is used in this study to conduct a series of simulations that highlight
the interactions between waves and sea level. These simulations enable the assessment of wave-induced circulation, the impact of currents on wave dynamics, and the mutual interactions between waves and currents. We conducted several simulations using the following configurations: W-0, an uncoupled wave model simulation; C-0, an uncoupled circulation model simulation; C-WF, where the wave model exclusively supplies fields to the circulation model; and W-C, the two-way coupled configuration. The outcomes of our simulations are then
compared with those reported in the existing literature, as cited in the discussion of the results.

The tidal inlet (Figure 2(b)) used in this study derives from the work of Cobb and Blain, (2002), based on Kapolnai et al. (1996). The domain recreates an inner basin (lagoon or embayment) measuring 20 km by 10 km, linked to an outer ocean basin of 49 km by 20 km on the continental shelf via a narrow 3 km long and 2 km wide channel. The bathymetry shows a linear slope from a depth of 14 m offshore to 5 m at the channel's entrance, with a
consistent depth of 5 m beyond the channel. The unstructured grid resolution is approximately 125 m within the channel and the surrounding area and 1 kilometer throughout the remainder of the domain.

The simulation conditions, as in Cobb and Blain (2002), are detailed as follows. The wave model is forced at the lateral open boundary with a wave field featuring a peak period of 10 seconds and a SWH of 1 m. The waves are set perpendicularly to the southern boundary of the domain. Two different boundary conditions are applied to the
offshore boundary depending on the type of simulation. If the circulation model is forced only with waves, a zero-gradient boundary condition is enforced along this boundary to simulate an unperturbed sea surface (in this region wave induced set-up/set-down should not be significant). When the circulation is forced with both waves and tides the offshore boundary elevation is modulated with specified M2 tidal component with 0.15 m of amplitudes and zero phase at the boundary.

In **Error! Reference source not found.** we present a snapshot of the results at the 48-hour timestep from the two uncoupled simulations W-0 (a) and C-0 (c), and the coupled W-C (b and d) simulations, with a magnification on the area of the inlet as indicated in Figure 2. The area of magnification is reduced in the plots showing the currents



to better appreciate the vectors' patterns. In the experiment W-0 (a) the wave field demonstrats an homogeneous SWH of approximately 1.2 m as it approaches the inlet, with a reduction along the boundary of the inlet and a
spread of around 60° in the lagoon, reaching a maximum SWH of about 0.8 m at the northern boundary. The W-C experiment (b) displays a markedly different and asymmetric pattern as shown in previous work as Cobb and Blain (2002). The maximum SWH in this domain exceedes 1.4 m, and at the mouth of the inlet, the SWH is higher than in the uncoupled experiment. The wave spread in the lagoon is larger, around 80°, and the maximum SWH at the northern boundary is lower than in the uncoupled simulation, reaching a maximum of 0.5 m. According to the
difference in spreading, we can assume that the limited spreading in W-0 is related to the reduced numerical diffusion known to affect WW3 high-order propagation schemes, as the one used here. Interestingly, the presence of currents in W-C appears to mitigate this phenomenon. The simulation C-0 (c) depicts the currents along the inlet, revealing four small meanders at the corners of the inlet and a central jet flanked by two eddies. Additionally, two large, nearly symmetrical eddies are observed in the lagoon. In the coupled simulation W-C (d), the four
meanders shift towards the center, resulting in the formation of two eddies between the meanders and the inlet walls. The central jet is less pronounced compared to the C-0 simulation, and the velocities at the inlet mouth, diverging outward, are more intense. This increased intensity contributes to the rise in significant wave height (SWH) observed in Figure 4(b). Inside the lagoon, the pattern of two large eddies is less pronounced, and a significant current is observed.


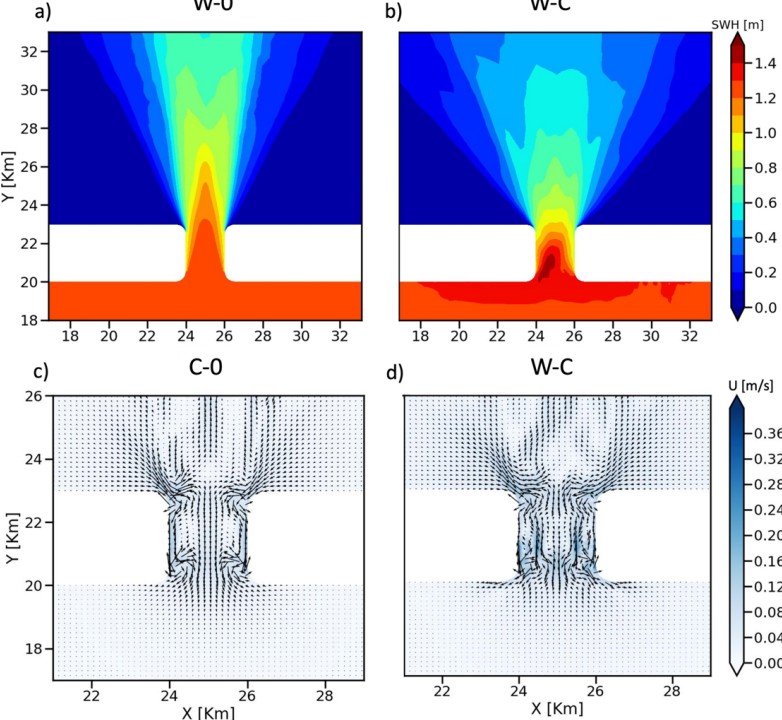

**Figure 4 Wave-forced coastal inlet at steady condition: significant wave height for the wave-only simulation (a) and for the two-way wave-currents coupled simulation (b); currents for the wave-to-current forced simulation (c) and for the two-way wave-currents coupled simulation (d).**

Following the work of Cobb and Blain (2002), we compared different simulations along the transect 1-2 (shown in Figure 2(b)). In the first validation at the 48-hour timestep, we qualitatively analyze the pattern of SWH from two configurations (Figure 5(a)). The W-0 configuration well describe the shoaling of waves entering the inlet and lagoon. The SWH exhibits a gradual increase to a maximum of approximately 1.1 m at 20 km from the southern boundary, followed by a reduction to 0.6 m at 30 km. The pattern observed in the coupled experiment
W-C alignes with W-0 up to about 17 km, where the waves begin to be influenced by the currents flowing outward



from the lagoon. This interaction induces refraction, resulting in steeper and higher waves that exceed 1.3 m. In the W-C configuration, the SWH breaks earlier, leading to a rapid drop to 0.6 m at 28 km.

**Error! Reference source not found.**(b) effectively illustrates the coastal processes of wave setup and setdown, as well as the complex interaction between currents and waves near the coast. In this plot, the water level is at rest and theoretically should remain constant at 0 m. The impact of wave forcing, namely the contribution of radiation stress to the circulation, is evident from the deviation of water elevation from the rest condition. A setdown correspondes well with the wave shoaling indicated by W-0 in plot a), reaching approximately -5 mm at 20 km from the southern boundary and +5 mm with a steeper slope at 33 km in experiment C-WF.

Wave setup and setdown are further amplified in the fully coupled configuration W-C, where current-induced refraction and breaking result in a higher momentum flux to the water. This configuration exhibits a negative peak of setdown at -2.5 cm and a positive peak of setup at +1 cm, consistently occurring at around 25 km.

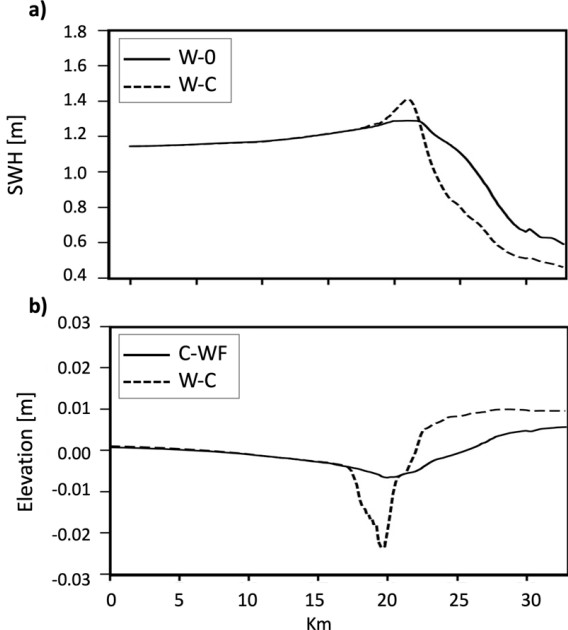

**Figure 5 SWH (a) and water elevation (b) of the transects along section 1-2 as indicated in Figure 2(b). The figures replicate the Cobb and Blaine (2002) simulations: in (a) we compared W-0 and W-C simulations, while in (b) we compared C-WF with W-C experiment.**

The last analysis of wave-current interaction in an idealized testcase examines the differences between the C-0 and W-C configurations within a tidally driven framework (Figure 6). During both flood (a) and ebb (b) phases, the smoother tidal profiles observed in C-0 are perturbed by the wave fields interactions in the W-C simulation. The difference between the configurations are slightly larger during the flood phase, around 1.5 cm, and less than 1 cm during the ebb phase of setdown. Both phases indicate a wave setup occurring beyond 22 km.





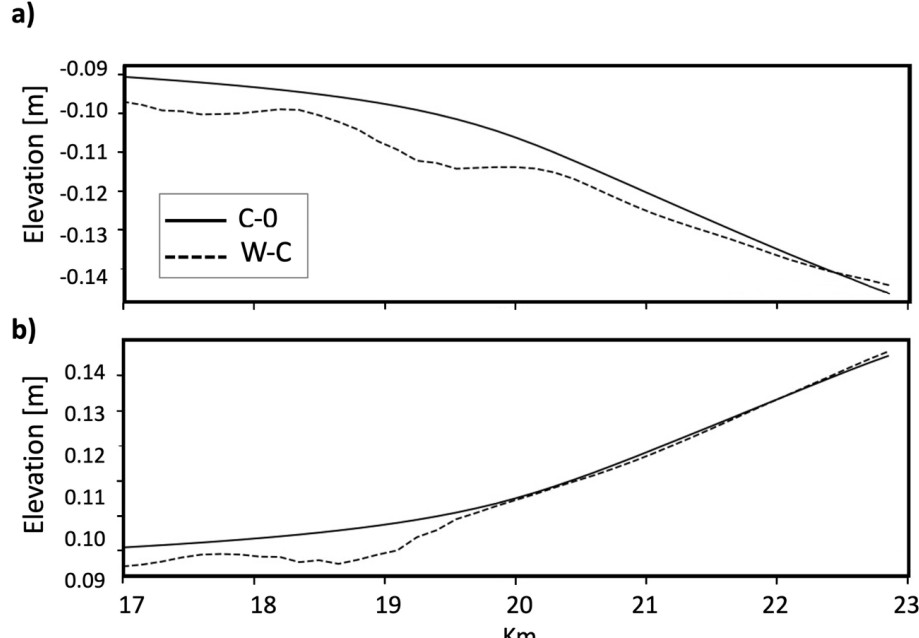

**Figure 6 Magnification (at 17-23 Km) of the water level along the transect 1-2 shown in Figure 2(b) for configuration C-0 and W-C in a tidally-driven framework at the ebb (a) and flood (b) phase.**

In all the qualitative validations (Figure 4, Figure 5, and Figure 6), the new coupled framework presented in this work produces accurate results that align with the expected physical behavior and are consistent with findings from other studies in literature (Cobb and Blain, 2002; Wu et al., 2011; Xia et al., 2020). This agreement is evident in both the spatial pattern and the range of variation for sea level and SWH, indicating that the model accurately captures not only the general shape and structure of the observed sea level and wave height patterns but also the

magnitude and fluctuations in their values.

### 4.   Simulation of circulation-wave fields during the Medicane Ianos

Medicane Ianos, is one of the most intense Mediterranean storms since the onset of satellite observation (von Schuckmann et al., 2022). It is notable for its duration and severity, resulting in significant rainfall, flooding, damage, and fatalities (Zekkos et al., 2020). It originated off the Libyan coast near the Gulf of Sidra on September 14[th], 2020, and moved toward the Ionian Sea and Greece between September 17[th] and 20[th]. The storm intensified as it moved northward and eastward, reaching peak strength on September 16[th] and passing through central Greece, particularly impacting the regions around Karditsa and Farsala. Ianos eventually weakened as it shifted southeast

towards Crete. With wind speeds reaching 110 km/h, the storm caused widespread flooding and destruction, leading to four fatalities in Greece and substantial damage, including the flooding of approximately 5000 properties in Karditsa alone (Henson, 2020).

### 4.1.   Set-up of the model

The numerical domain, shown in Figure 7, encompasses the entire central South Mediterranean Sea, with three open lateral boundaries: in the Otranto Channel to the North, the Sicily Strait to the West, and the Ionian Sea (between Greece and Libya) to the East. The domain is discretized with an unstructured grid, featuring a horizontal



resolution of approximately 2 km offshore, 1 km along the coasts, and 50 m in the area where Medicane Ianos made landfall in Greece, including the islands of Zakynthos and Kefalonia.

The circulations and wave models are forced at surface with atmospheric fields from ECMWF analysis data (10 km resolution and 6h frequency). The surface fields are corrected along the coastal boundaries following the sea-over-land procedure described in Kara et al. (2008). The atmospheric variables used for the circulation include 2 m air temperature, 2 m dew point temperature, total cloud cover, mean sea level atmospheric pressure, and total

precipitation. Meridional and zonal 10 m wind components are used for both wave and circulation models.

The modelling approach is based on the downscaling of Copernicus Marine Service[1] products released at the Mediterranean Sea regional scale. The circulation model is three-dimensionally nested into MEDSEA_MULTIYEAR_PHY product (Table 2) both in terms of initialization and open boundaries. The scalar fields (sea level, temperature, and salinity) are treated through a clamped boundary condition, while the velocity

fields are imposed as nudged boundary condition using a relaxation time of 3600 s.

The wave model is initialized using a parametric fetch-limited spectrum based on the initial wind field and is forced at the lateral open boundaries by the MEDSEA_MULTIYEAR_WAV product (Table 2). Mean wave parameters, such as significant wave height, mean wave direction and peak period, are used to reconstruct the wave spectra, following the approximation described by Yamaguchi (1984).


**Table 2 Datasets used in this study**

| Datasets | Provider | Usage | Web site |
| --- | --- | --- | --- |
| MEDSEA_MULTIYEAR_PHY | Copernicus Marine Service | Circulation model lateral boundary | https://data.marine.copernicus.eu/product/MEDSEA_MULTIYEAR_PHY_006_004/description |
| MEDSEA_MULTIYEAR_WAV | Copernicus Marine Service | Wave model lateral boundary | https://data.marine.copernicus.eu/product/MEDSEA_MULTIYEAR_WAV_006_012/description |
| SST_MED_SST_L4_REP_OBSERVATIONS | Copernicus Marine Service | SST anomaly evaluation | https://data.marine.copernicus.eu/product/SST_MED_SST_L4_REP_OBSERVATIONS_010_021/description |
| WAVE_GLO_PHY_SWH_L3_NRT | Copernicus Marine Service | Wave validation | https://data.marine.copernicus.eu/product/WAVE_GLO_PHY_SWH_L3_NRT_014_001/description |
| Katakolon tide gauge | Permanent Service for Mean Sea Level | Sea level validation | https://psmsl.org/data/obtaining/stations/1240.php |
| EMODnet bathymetry 2020 | EMODnet | Model bathymetry | http://www.emodnet-bathymetry.eu. |
| IFS010 | ECMWF | Atmospheric forcings | https://www.ecmwf.int/en/forecasts/documentation-and-support/changes-ecmwf-model |

---

[1] https://marine.copernicus.eu/it



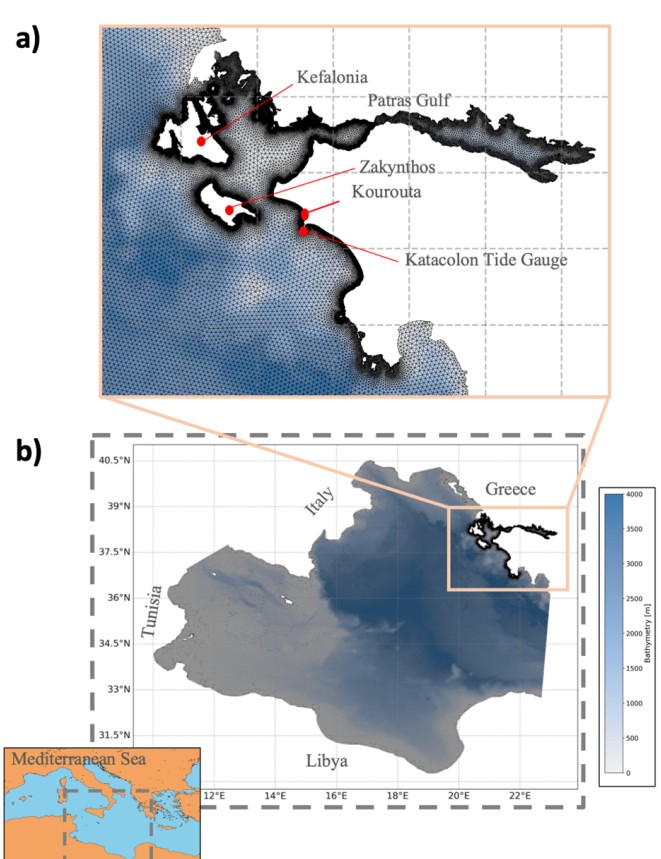

Figure 7 Spatial unstructured-grid and bathymetry for the Ionian Sea (b) used for the Ianos Medicane simulation, with an enlarged view on refined areas of affected area (a).

### 4.2 Results and discussion

The Medicane Ianos reached its maximum intensity at 18:00 on September 16[th], 2020, as it entered the Ionian Sea (Figure 8). The coupled modeling framework provides a comprehensive analysis of the event, revealing the storm's extensive impact on the wave field across the central Mediterranean Sea. The affected area spans over 5°x 5°, with significant wave heights averaging over 3 m and reaching up to 6 m in the eyewall region (Figure 8(a)). The Medicane's influence on sea level is clear, as a pronounced positive sea level anomaly exceeding +20 cm is observed, particularly within the storm's core (Figure 8(b)). This anomaly stands out against the low-tide conditions, providing a distinct footprint of the storm. Although sea surface temperature do not exhibit a consistent pattern at the storm's peak, cooler waters are noticeable in the southeastern region of Sicily (Figure 8(c)). Additionally, the sea surface currents display a well-defined pattern, with velocities exceeding 1 m/s along the eyewall and a divergent flow branching northeastward and southeastward (Figure 8(d)).



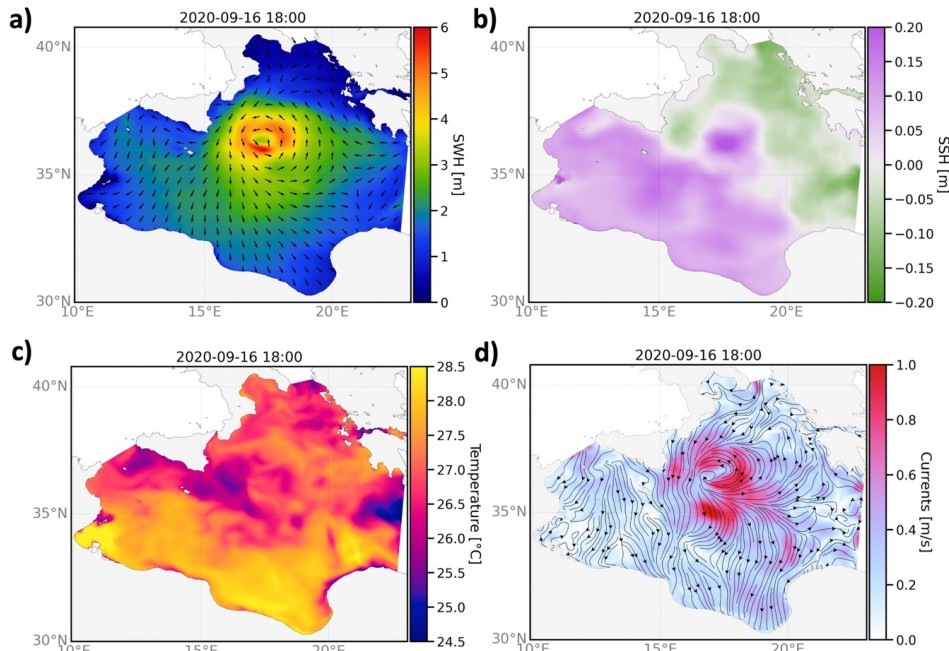

**Figure 8 Maps of significant wave height (a), sea level (b), sea surface temperature (c) and surface circulation (d) on 2020-09-16 18:00 (peak time in open ocean) for the entire domain from two-way coupled simulation.**

At 06:00 on the 18[th] of November, the Medicane made landfall in Greece, impacting the islands of Zakynthos and Kefalonia, as well as the Gulf of Patras (Figure 9). The SWH simulated during this event, as shown in Figure 9(a), is notably high for this region of the Mediterranean, reaching up to 4.5 m along the western coastlines of the islands. The cyclonic circulation associated with the Medicane intensifies the waves on the northern part of the Gulf of Patras, which is particularly affected due to the nearly perpendicular wave direction. Furthermore, the confluence of currents from the north and south of Zakynthos (Figure 9(b)) resultes in current speeds reaching up to 1.2 m/s in this convergence zone. The combined impact of these high waves and strong currents lead to a significant storm surge, exciding 30 cm, in the northern part of the Gulf of Patras (Figure 9(c)).

In the following sections, we investigate the effect of the coupling on the simulation of the Medicane.





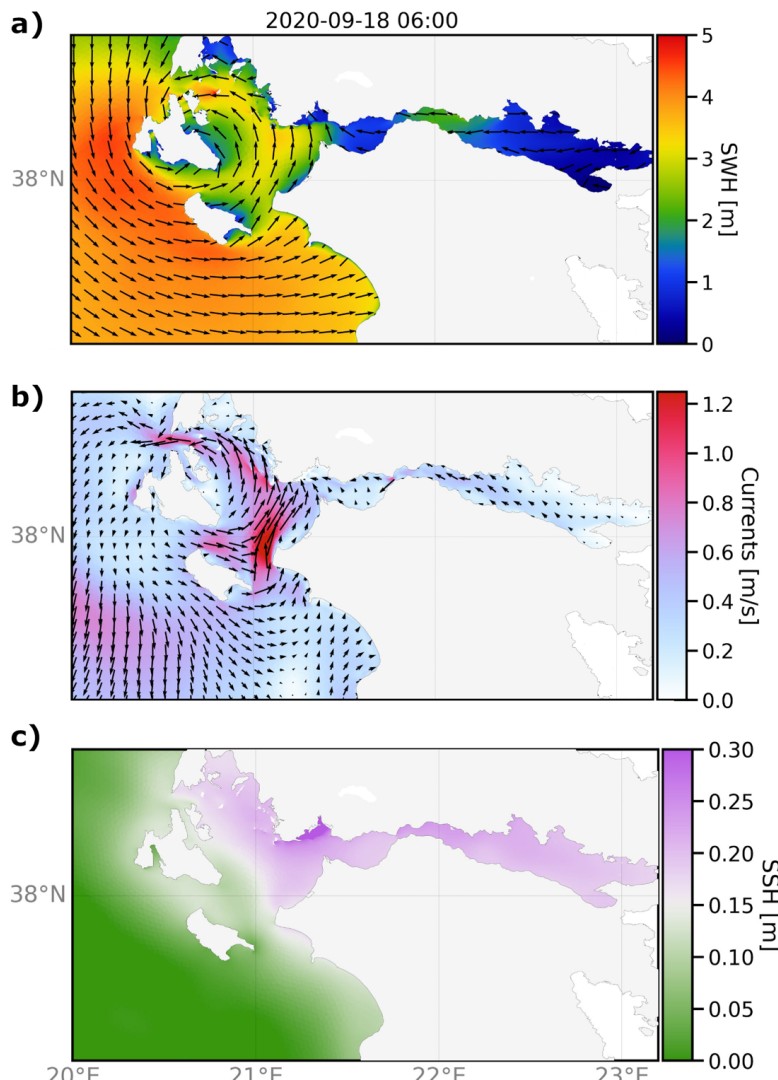

**Figure 9 Maps of significant wave height (a), surface circulation (b) and sea level (c) on 2020-09-18 06:00 (peak time in coastal zones of Greece) for the Greek coastal areas from two-way coupled simulation (W-C).**

### 4.2.1. Effects of circulation on waves

The validation of the wave model results and the difference between the W-C and W-0 configurations are shown in this paragraph using two approaches. In **Error! Reference source not found.** we compare the simulations with the satellite altimeter tracks (Table 2) which crossed the Medicane. In details, three altimetric measurements from Jason-3, SARAL-Altika and Sentinel-3B are available on the 17[th] September, and the SWH results from the W-C configuration are overlapped with the satellites measurement and shown in **Error! Reference source not found.**(a),(b),(c) respectively. The model simulates the extreme event with a very high quality, and the patterns of the modeled wave field match with good agreement the satellite data. For Jason-3 and Sentinel-3B the measured



SWH reaches 5 m, while for SARAL-Altika, the SWH is slightly lower because the track is slightly eastward to the Medicane. Figure 10 panels (d), (e), and (f) illustrate the along-track validation for both the W-C and W-0 configurations. The difference between the two configurations can be a few tens of centimeters in some cases, while in others, the time series overlap. However, the W-C configuration demonstrates a reduction of
approximately 3-4% for the Root Mean Squared Error (RMSE) for Jason-3 (d) and Sentinel-3B (f) tracks. It is worth mentioning that, although this analysis shows better peak predictions for the W-0 configuration, the W-C configuration more closely matches the observations overall, both during growth and decay phases. A potential motivation for the reduction of SWH at peaks in the coupled experiment is due to the alignment of currents and wind, which reduces wind stress. This is attributed to the fact that the relative wind speed is lower than the absolute
wind speed.

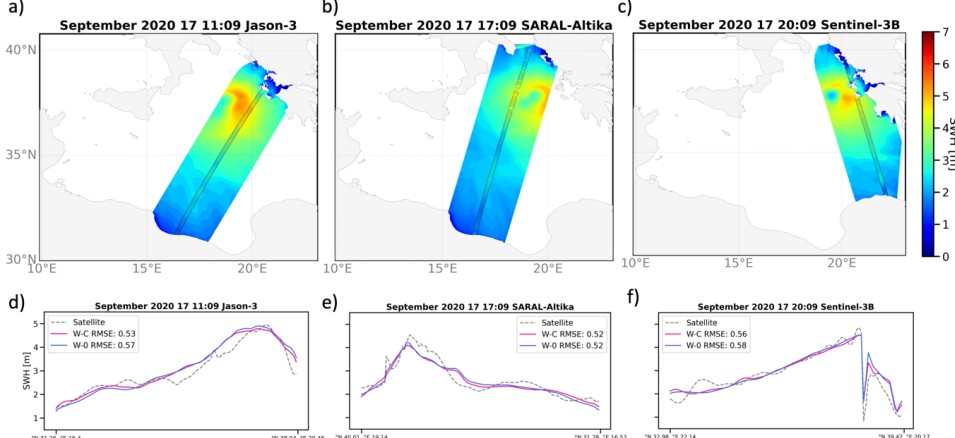

**Figure 10 Maps of SWH against satellite altimeter (Jason3, Saral-Altika, Sentinel-3b) tracks (inset circles) for the W-**
**C configuration (a) (b) (c). SWH along-track comparison between the two model configurations and satellite tracks is reported in (e), (f) and (g).**

For a more robust assessment of the wave model, we analyze the time window including the days the Medicane Ianos persisted over the central Mediterranean Sea, from September 15$^{th}$ to 19$^{th}$ (Figure 11). From this investigation, the model exhibites a very high accuracy in representing the extreme event, with a correlation
coefficient of 0.95, a negative bias of approximately 20 cm, and an RMSE of 33 cm. The W-C / W-0 comparison shows a reduction in bias of around 6% and a decrease in error of approximately 3% in the coupled configuration, confirming the previous findings of Causio et al. (2021).



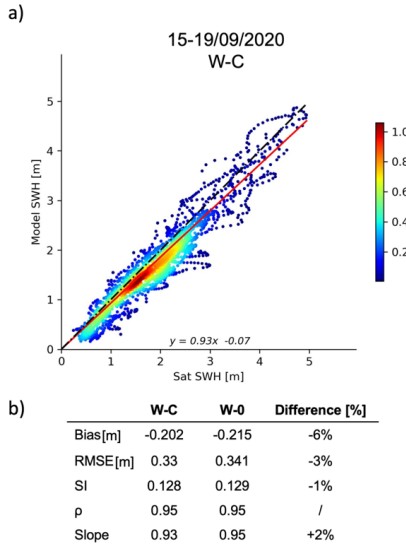

**Figure 11 (a) SWH scatter plot for the W-C configuration is shown, with colors indicating the plot density of occurrence. The black line represents the best-fit line, while the red line denotes the linear regression of the data. (b) Statistics (Root Mean Squared Error – RMSE, Bias, Scatter Index – SI, Pearson's correlation coefficient – ρ, and the slope of the fit) are summarized and expressed in meters, providing comparisons for the W-C and W-0 configurations as well as the percentage difference between them.**

### 4.2.2. Effects of waves on circulation

In Figure 12 we illustrate the drag coefficient ($Cd$) from the W-C (Figure 12(a)) and C-0 (Figure 12(b)) configurations at the peak of the event (16[th] September 2020 at 18.00) and their differences (Figure 12(c)). The $Cd$ from the uncoupled simulation, which is derived from the Hellerman and Rosenstein (1983) formulation, exhibits a more homogeneous field and overall higher values with respect to the coupled system, except in the area of the Ianos passage at the peak time, where the coupled simulation is able to provide larger $Cd$ values. Significant differences between the sea-state dependent drag (W-0) and the Hellerman and Rosenstein (1983) formulation (used in C-0) occur in regions with lower values of significant wave heights (see Figure 8(a)), particularly in the Northern Ionian Sea, the eastern part of the study domain, and the Gulf of Gabes. Additionally, the wider range for $Cd$ allows to reach higher values; the C-0 configuration never exceeds the value of $2.5 \times 10^{-3}$, while the W-C configuration goes beyond $3.5 \times 10^{-3}$. In both configurations, the pattern of extreme drag is asymmetric, with the southern part of the Medicane experiencing the highest values.



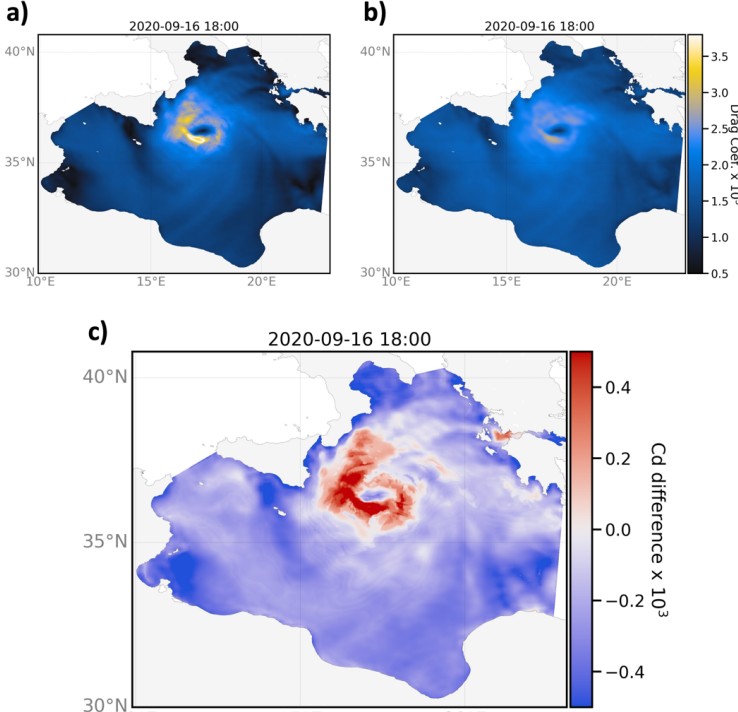

**Figure 12 Maps of drag coefficient (Cd) at the peak event in open ocean (2020-09-16 18:00). (a) Drag coefficient computed by WW3 in the W-C experiment. (b) Drag coefficient computed via Hellerman and Rosenstein (1983) in the C-O experiment. (c) Difference between (a) and (b).**

The passage of a strong cyclone over a water body induces surface cooling due to both upwelling and increased evaporation (Price, 1981). In Figure , we present the Sea Surface Temperature (SST) footprint induced by Ianos' passage over the central Mediterranean Sea, following the approach of Clementi et al. (2022). The footprint results from the comparison of the SST of the day after the Medicane hit Greece (19th September) with the SST of the day before Ianos' formation (14th September). The footprint is computed for L4 gridded satellite observations (Table 2) (Figure (a)), W-C configuration (Figure (b)), and C-0 configuration (not showed). Additionally, the difference between the footprints of W-C and C-0 is shown in  Figure (c). Specifically, for the satellite data, the foundation temperature is used, which approximately corresponds to the sea temperature during the night-time at a depth of around 2-5 m. For this reason, in this analysis we used the model temperature at midnight at a depth of 3 m.

The SST footprint from satellite shows a general cooling of the central Mediterranean Sea, with the exception of the southern coasts, and two distinct areas of larger significant cooling. The southern Ionian Sea, where the Medicane experiences its highest intensity, shows the largest cooling, dropping the temperature down at –5°C. Here the footprint follows the Medicane track with a northeast orientation.

A qualitative comparison between the satellite data and the W-C and C-0 configurations illustrate that both configurations exhibit adequate patterns but show a stronger impact on SST compared to the observations. However, the W-C configuration demonstrates slightly warmer behavior (Figure (c)), and the footprint's width is narrower than in C-0, more closely matching the pattern observed in satellite data. These differences can be attributed to the sea-state dependent drag coefficient which modifies the circulation in the area.

Furthermore, Figure 13 shows the path of the medicane as it passed through the Ionian Sea (from Libya at point 1 to Greece at point 17), highlighting the location of the sea level pressure minimum every 6 hours.



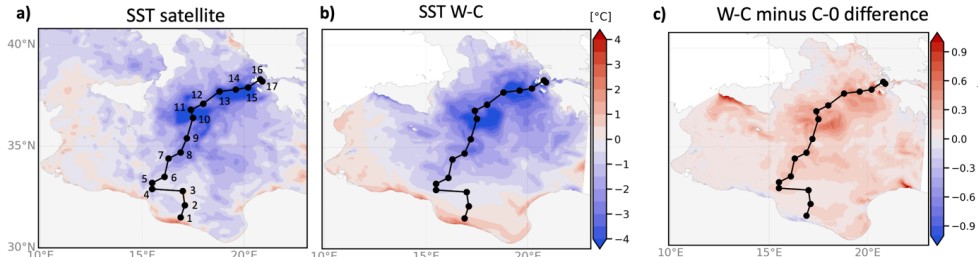

**Figure 13 Sea surface temperature anomaly [°C] between 19th Sept. (after Ianos) and 14th Sept. (before Ianos) from satellite (a ), two-way coupled model W-C (b), and SST difference between the anomaly of W-C and C-0 (c). The black path represents the Medicane Ianos path identified as minimum sea level pressure location every 6 hours. In panels (a) and (b) the colorbar range is set to [-4, 4 °C] to better highlight the footprints pattern, even though the difference reached -5 °C.**

Consequently, we analyzes the vertical profile of model temperature along the Medicane track (Figure ) for both the W-C and C-0 configurations. In this assessment we compute, as in Figure , the temperature anomaly between 19th and 14th September, and then, the difference between W-C and C-0. The range of temperature anomaly spans from approximately -0.6 to 0.6 °C. Notably, the largest difference between the two configurations occurs in the Ionian Sea, particularly from point 10 onward. Throughout the track, the W-C configuration is warmer in the mixed layer, being the latter generally slightly shallower in the coupled simulation, with a few exceptions at points 3 and 14 where the uncoupled configuration exhibites a slightly thinner mixed layer. As Medicane Ianos moves towards Greece, the plot indicates that the W-C configuration footprint is even warmer in the mixed layer than C-0, and even colder below the pycnocline, suggesting a strenghtening of the stratification in W-C. Point 13 and 14 are peculiar because they show warmer W-C even below the mixed layer up to 60 m. This difference can be attributed to the less intense upwelling observed in the W-C configuration, which results in smaller cold wedges (located at points 10-12 and 13-15 in Figure ) compared to those in the C-0 configuration.

This suggests indicates that the impact of this coupling on the vertical structure is primarily confined to the uppermost 50 m in the central Mediterranean Sea, nevertheless in some cases, as in coastal areas, the impact extends deeper, reaching up 100 m and beyond.



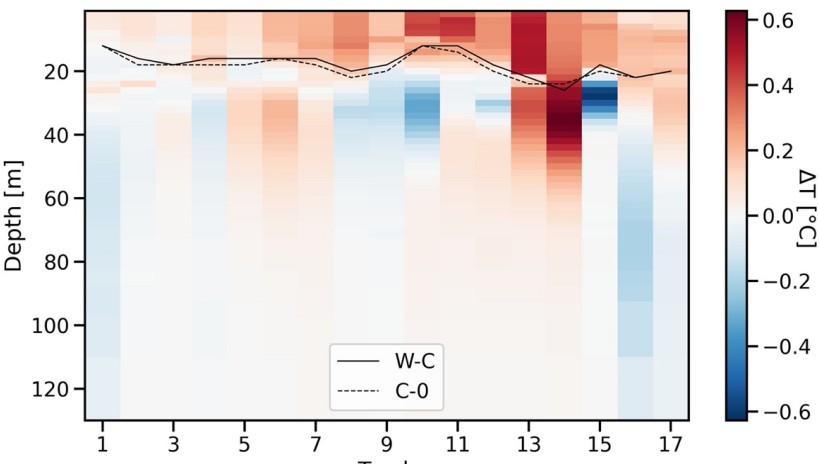

**Figure 14 Anomaly Temperature difference between W-C and C-0 for the Medicane Ianos passage over the central Mediterranean Sea. Black lines refer to the mixed layer depth for W-C (solid line), and C-0 (dashed line). The x-axis corresponds to the locations shown in Figure13(a).**

One of the main contributions expected from this coupled modeling framework is the enhanced capability to predict storm surges, by inclusion of the radiation stresses. Medicane Ianos was chosen for this study because it made landfall close to the Katakolon tide gauge, which provides a suitable reference for comparison between the sea level observations and the W-C and C-O configurations (Figure ). The C-0 configuration demonstrates good accuracy in predicting sea level at the tide gauge location; however, the W-C configuration represents more accurately the storm surge induced by the Medicane (the event is highlighted in yellow in Figure (a)). Both configurations show nearly identical performance under normal conditions (before and after the event). The maximum difference between the two configurations is 4 cm, allowing the W-C configuration to simulate more precisely the two peaks of the storm surge. Notably, the inclusion of wave effects lead to a more rapid drop in surge after the storm passage, improving also the fit for the first tide trough after the storm (the third tide trough within the yellow area in Figure (a)). It is worth mentioning that the wave contribution to the Ianos storm surge is approximately 10% at the peak of the event, although this value can be significantly higher in other situations. To illustrate this, Figure (a) bottom panel shows the same time series comparison for a location few km far from the tide gauge, along the nearby beach of Kourouta. The plot reveals that the waves provide a larger contribution in this location, resulting in a surge about 11 cm higher than the one showed in C-0, accounting for roughly 30% of the total contribution. The difference between the timeseries, as shown in Figure 15(c), can be attributed to the wave field direction, which makes the tide gauge partially sheltered, while the beach faces more orthogonal waves. The significance of wave direction is further emphasized along the northern shoreline, where the wave setup is even more intense due to the waves being nearly orthogonal to the coast.

Notably, the wave contribution to the storm surge during the Medicane Ianos is comparable in percentage to findings of Dietrich et al. (2011) for Katrina and Rita hurricanes in the Gulf of Mexico, and Kim et al. (2010) during Typhon Anita in South China Sea.



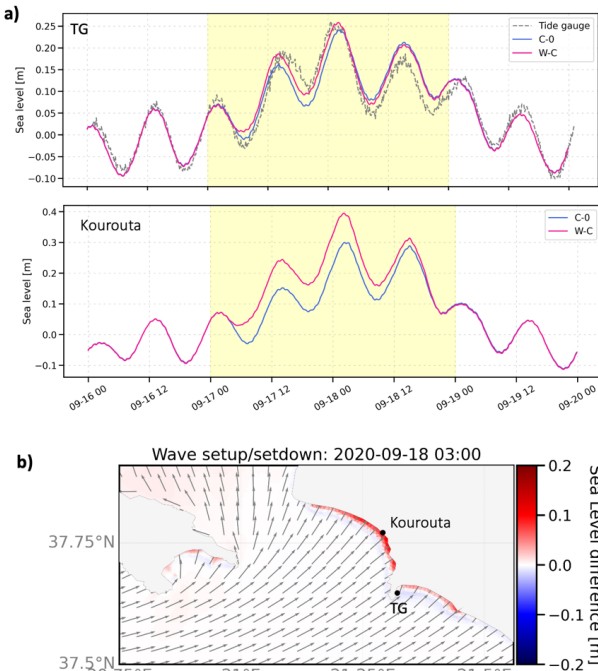

**Figure 15 Timeseries of TWL for C-0 and W-C configurations. In plot (a) upper panel, the model are compared with observed data at Katakolon tide gauge -TG (in dashed gray). In the plot (b) bottom panel, the model configurations are compared at the Kourouta beach. The shaded yellow area highlights the two days impacted by the Medicane induced surge. Plot (b) highlights the wave setup (in red) and setdown (in blue), computed as the TWL difference between C-0 and W-C. Additionally, it shows the location of the timeseries (in plot a), as well as the mean wave direction represented by vectors.**

### 5. Conclusions

This study represents the first two-way coupling of the SHYFEM-MPI model with WW3 using an external coupler, marking an advancement in the simulation of storm surges during extreme events like Medicane Ianos. Our primary goal is the evaluation of storm surge dynamics, nevertheless, together with radiation stress, we incorporated also additional coupled physical processes, such as sea-state-dependent wind stress, doppler shift, sea-level-dependent water depth for wave physics, and the inclusion of relative wind based on air stability parameters. Rigorous validation through well-established idealized test cases, including the tidal inlet and planar beach scenarios, confirm the reliability of our coupled modeling framework.

The real-case simulation and comparison of coupled versus uncoupled configurations highlight the added value of incorporating the wave component in storm surge simulations, particularly for coastal applications. Notably, during Medicane Ianos, wave setup contributes to approximately 10% of the sea level variation at the Katakolon tide gauge, with a significantly greater impact—around 30%—in nearshore areas. The inclusion of sea-state-dependent drag coefficient introduces a larger dynamics to the simulations, with a lower drag in areas distant from the storm and increases drag near the eyewall of the Medicane.

Our findings indicate that the coupling effects are more pronounced in the northern part of the Medicane's path, where the storm intensity is largest. The coupling leads to a slightly thinner and warmer mixed layer and, in some locations, affects the water column up to 100 m, likely due to reduced upwelling of deeper waters. The impact of circulation on wave dynamics is appreciable, as coupling reduces bias by approximately 6% and overall error by around 3% in the wave model.



The numerical framework developed in this study proves to be an effective tool for simulating both large-scale and coastal processes, with a particular strength in describing storm surge dynamics. This system holds significant potential for enhancing coastal hazard identification, early warning systems, and the management and planning of sea and coastal activities. The encouraging results from this work lay the foundation for further development, including the integration of additional processes such as Stokes drift and sea-state-dependent vertical mixing, which are currently neglected or parameterized in this version of the circulation model.

## 6. Authors contribution

SC contributed to conceptualization, data curation, investigation, formal analysis, methodology, validation, visualization writing the original draft, and reviewing and editing.
SS contributed to data curation, validation, visualization, software and writing – review & editing.
IF contributed to conceptualization, formal analysis, supervision, writing – original draft preparation
GDC contributed to data curation, methodology, visualization and writing – review & editing.
EC contributed to formal analysis, methodology, and writing – review & editing.
LM contributed to formal analysis, methodology, investigation, visualization and writing – review & editing.
GC contributed to formal analysis, funding acquisition, resources and project administration.

## 7. Acknowledgements

This work has been funded by:
- EDITO-Model Lab. Project number: 101093293
- PNRR-HPC. Spoke 4 - ICSC–Centro Nazionale di Ricerca in High Performance Computing, Big Data and Quantum Computing, funded by European Union–NextGenerationEU; Project number: CN00000013; CUP: C83C22000560007
- EOatSEE project. Subcontract DME-CMCC Ref. DME-CP51 no. 2022-008

whose support provided the necessary resources for the research conducted in this study.

## 8. Competing Interest

The authors declare that they have no conflict of interest.

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
