# Peer review of "Coupling ocean currents and waves for seamless cross-scale modeling during Medicane Ianos"

_EGUsphere, 2024_

## Author Response (AR1)

**Reviewer1**

Review of "Coupling ocean currents and waves for seamless cross- scale modeling during Medicane Ianos"

This is an interesting paper. It shows the impact of coupling a wave model to the circulation model SHYFEM and is presented as a novel feature. It might well be for SHYFEM but note that from a scientific point of view, this is not a novel feature. Over the last 30 years, there have been many studies recommending adding wave effects to circulation models. Some have led to operational implementations. Nevertheless, I would still recommend the publication of this research in the framework of research to develop systems to improve the modelling of extreme weather impact.

Comments

Line 70: I will note the very relevant study of Ferrarin et al. (2023):

https://nhess.copernicus.org/articles/23/2273/2023/

- **Thanks for the suggestion, we included the study also in this part of the manuscript**

Line 142 and Line 187: which Txxx package (T471?). Because you are using ST4, you need to mention that Janssen original formulation was modified by Ardhuin et al. (2010) of account for sheltering effect of long waves on the short waves (which Txxx package, hence which sheltering coefficient and hence the possible range of variation of betamax. It has not been mentioned which actual Txxx package was use and/or which value of betamax was selected. Users of WW3 tends to wrongly assume that it is ready out of the box to be coupled with other earth system model components. However, when using ST4, they then play around with betamax until they get satisfactory results. But, this has implication on the strength of the momentum flux. It is therefore imperative that it is demonstrated that the drag coefficient hence obtained is within physical bounds as expected from observations. For instance, I would strongly encourage comparing the drag coefficient ($C_d (U_{10})$) obtained with the coupled system to Edson et al. (2013) as it seen as one of the best estimates of the drag coefficient from the field

This would nicely complement Figure 12 and will inform reader on how it compares to Hellerman and Rosenstein (1983) used by the authors when not coupled.

https://journals.ametsoc.org/view/journals/phoc/43/8/jpo-d-12-0173.1.xml

with the corrigendum

https://journals.ametsoc.org/view/journals/phoc/44/9/jpo-d-14-0140.1.xml

- **Thank you for pointing this out. We have clarified in the revised text that our simulations were performed with BETAMAX=143, which corresponds to the** T471 package**. This parameter has now been explicitly mentioned at line 159.**

- **Drag Coefficient Comparison (Edson et al., 2013)**
  As suggested by the reviewer, we compared the both drag coefficients from Hellerman and Roseintein, and the one from WW3 to the observations-based formulation by Edson et al. (2013). We are pleased to report that the results obtained using our wave model are in good agreement with the latter, as showed by the difference in the following Figure. This ensures the physical validity of the calculated drag coefficient. However, we would not prefer to include an additional figure in Figure 12 at this stage, as it would introduce another Cd formulation, which could require further comparisons with other drag coefficient parameterizations. Instead, we provide the comparison results in this response.

[Figure]

Line 165: With only 6-hourly wind forcing from ECMWF, I can accept that updating the wave information only every hour was found to be optimal. Ideally with such an intense system such as a medicane, it would ideally be better to use higher frequency forcing (at least hourly), as would be available from short range forecasts. In that case, I would question whether better results could be obtained with more frequent updates in the coupling.

- **Thank you for the comment. Unfortunately, we do not have higher frequency wind forcing available for this hindcast. However, we have investigated the impact of increasing the coupling frequency in idealized test cases. Specifically, we tested coupling intervals of 5, 10, 30, and 60 minutes but did not observe significant improvements in storm surge evolution. Considering these findings and the computational cost associated with running the coupled system at higher frequencies, we opted for a 1-hour coupling interval, as it provided satisfactory results in the idealized scenarios.**

Line 429: A bit unclear here. IfI assume that you have use the winds relative to the surface currents (i.e. ECMWF winds – surface currents in vector calculation), but ECMWF analysis winds have been produced without the knowledge of those currents. Hersbach and Bidlot (2009)

have shown that had the ECMWF analysis known about the currents, then the whole wind profile in the boundary layer would have changed to account for the change in surface stress. They have estimated that on average, only half of the surface currents should be subtracted from the analysis 10m winds

Hersbach H. and J.-R. Bidlot, 2009: The relevance of ocean surface current in the ECMWF analysis and forecast system. Proceeding from the ECMWF Workshop on Atmosphere-Ocean Interaction, 10-12 November 2008.

**However**, you might have meant the application of (7). If it is so, I would strongly recommend that you call this impact of gustiness on wave growth or something similar as relative winds usually means what I have described above.

- **Thanks for pointing this out. We mistakenly defined 'relative wind' instead of 'effective wind' in the conclusions. This has been corrected in line 214**

There is no separate investigation of the actual impact of this wind adjustment (7) on the simulation. It is presented as a novel contribution and so I would expect to see its impact.

- **We thank the reviewer for pointing this out. We have revisited the statement to clarify that this represents the first implementation of this formulation in the SHYFEM-WW3 coupling. The revised sentence now reads:**

  *'Present work incorporates new processes into the two-way coupling between SHYFEM- WW3, including the sea state dependent momentum flux and wind field correction based on the air stability parameter. The study marks the first study to implement a three-dimensional baroclinic version of wave coupled SHYFEM model. '*

  **Additionally, we included in the thext at lines 463-466 the sentence:**

  *"In this study, we do not present through attribution simulations the explicit quantification of contribution of the air-stability parameter and currents. This decision is based on our prior investigations, which have already demonstrated its effectiveness and role (Clementi et al., 2017; Causio et al., 2021)"*

  **To refer to our previous papers where we have already applied and validated and detailed this wind adjustment (Clementi 2017 and Causio 2021), which is why we chose not to reiterate its contribution in this paper."**

Line 419: The authors state that the main contribution of the waves to the storm surge by the inclusion of the radiation stresses. However, several studies have also found that the sea state dependent momentum flux could also be a relevant factor (Bertin et al. 2015, Pineau-Guillou et al., 2020). It would be good to know the impact of either wave effect on the storm surge for Ianos

Bertin, X., Li, K., Roland, A., Bidlot, J.-R. 2015: The contribution of short-waves in storm surges: Two case studies in the Bay of Biscay. Continental Shelf Research 96, 1-15.

Pineau-Guillou L., Bouin M. N., Ardhuin F., Lyard F., Bidlot J. R., Chapron B., et al. (2020). Impact of wave-dependent stress on storm surge simulations in the North Sea: Ocean model evaluation against in situ and satellite observations. Ocean Modelling 154, 101694. doi: 10.1016/j.ocemod.2020.101694

- **We previously investigated the impact of variations in the Cd formulation on sea level changes during another extreme event in the Mediterranean Sea, the Medicane Helios (February 2023). Our findings indicated that Cd, and thus the air-sea momentum flux, had no significant contribution to the storm surge. For coinciseness, that study employed the same coupled system as presented in this work. In that case, we observed that the surge was predominantly driven by radiation stress. To provide further clarification, we have included the following plot from that study, which demonstrates that the simulations (CD+SXY) and (SXY) produced very similar results, whereas the simulation using only CD yielded different outcomes (Note: SXY refers to the total radiation stress). Therefore, given the associated computational cost, we did not perform a dedicated attribution simulation, as our prior experience suggested that it would not provide additional valuable insights.**

[Figure]

**Minor corrections:**

Line 13: one of the most intense cyclones occurred -> one of the most intense cyclones to have occurred

- **Fixed**

Line 23: sea-state dependent momentum -> sea-state dependent momentum flux

- **Fixed**

Line 98: hurricane – medicane

- **Fixed**

Line 110: studied -> studies

- **Fixed**

Line 149: log-spatially varying wavenumber -> log varying frequency discretisation

- **Fixed**

Line 184: in (1), delta not defined

- **Fixed**

Line 185: wind-neutral -> neutral wind

- **Fixed**

Line 190: modified by the wave-supported stress $\tau w$ -> modified by the ratio of the wave-supported stress $\tau w$ to the total stress $\tau$. $\tau w$ represents the flux of momentum that is directly input into the wave field by the wind.

- **Fixed**

Line 192: according to WW3 manual, alpha_0 is most likely 0.0095 (one needs to know the exact Txxx version that was selected, or did you actually use 0.01?)

- **Thanks for finding this we fixed it.**

Line 201: "Abdalla and Bidlot (2002) formulated …" I know that the WW3 documentation for v 6.07 mentioned that it came from that work. But this is incorrect, ECMWF is using the gustiness parameterisation and forcing its wave model with neutral 10m winds, but it is not based on (7). I believe Hendrik Tolman proposed (7) but I can't find a reference. If the authors know where (7) came from, please do change the reference, or state instead "Abdalla and Bidlot (2002) inspired …" and quote WW3 because it is the option controlled by STAB3 switch?

- **Fixed**

Line 206: symbol h in Uh is confusing, since h has just been used for water depth. I recommend using U_z instead

- **Fixed**

Line 245: what spectral shape is imposed at the boundary with SWH=2m and Tp=10 s

- **We included the JONSWAP shape at line 250**

Same question for line 270

- **We included the JONSWAP shape at line 277**

Line 248: derives -> derive

- **Fixed**

Line 280 and 314: In Error! Reference source not found

- **Fixed**

Line 285: why is the W-0 not totally symmetric with respect to the axis of the inlet (line 1-2 in Figure 2)? (grid not fully symmetric with respect to that axis and/or the direction discretisation in WW3 does not align with the said axis ?)

- **The not-perfect simmetry in W-0 is related to the fact that the grid is not perfectly aligned with the propagation, thus the discretization and plotting in triangles created this slight diffence in the simmetry.**

Line 290: in the W-C simulation, the presence of non-uniform currents will cause wave refraction, therefore enhancing the spread as observed in the lagoon, both physically but also numerically.

- **We included at line 294 the sentence:**

  *"we can assume that it is caused by wave refraction induced by non-uniform currents, which enhances both physical anc numerical spreading. On the other hand, W-0 exhibits reduced numerical diffusion which is a characteristic of the WW3 high-order propagation schemes"*

Line 312: Could illustrate that indeed breaking intensity is increased. It is not sufficient to just state that the waves are becoming steeper, you have to show that they are actually breaking.

- **The sentence has been rephrased at lines 324-326 as:**

  *"This interaction causes refraction, resulting in steeper and higher waves that exceed 1.3 m around 18 Km. As result, the waves begin to break earlier and more intensely, leading to a rapid drop of SWH to 0.6 m at 28 km."*

Figure 6: Km -> km

- **Fixed**

Line 361: forced at surface with -> forced with

- **Fixed**

Line 384 and Figure 8 caption: 18:00 -> 18:00 UTC

- **Fixed**

Line 415 and 418: Error! Reference source not found

- **Fixed**

Line 416: Table 2 -> WAVE_GLO_PHY_SWH_L3_NRT, Table 2

- **Fixed**

Line 438: what is the collocation criteria between model and altimeter (closest in space and time (what is the output frequency of the model simulation))?

- **The sentence *"The model outputs are saved at 1-hour frequency."* Has been added at line 393.**
  **The sentence *"This analysis is based on the nearest point between observation and model in both time and space dimensions"* Has been added at line 444.**

Line 444: an RMSE -> a RMSE

- **Fixed**

Line 457: The sea state dependent Cd is connected to the state of development of the wave field and not directly the the significant wave height. So the comparison of high Cd should be made with wave age and not SWH.

- **The sentence has been rephrased at line 483 as: *"Significant differences between the sea-state dependent drag (W-0) and the Hellerman and Rosenstein (1983) formulation (used in C-0) occur in regions farther from the Medicane."***

Line 467: In Figure -> in Figure 13

- **Fixed**

Line 478: down at -5 -> down by -5

- **Fixed**

Line 493: Figure -> Figure 14

- **Fixed**

Line 494: Figure -> Figure 13

- **Fixed**

Line 499: exhibites -> exhibits

- **Fixed**

Line 505: This suggests indicates that -> This suggests that

- **Fixed**

Lines 516 and 518: Figure -> Figure 16

- **Fixed**

Line 566: add these references for Stokes drift and/or wave effect on upper ocean mixing. Also add wave effect on bottom friction.

Øyvind Breivik, Kristian Mogensen, Jean-Raymond Bidlot, Magdalena Alonso Balmaseda, Peter A.E.M. Janssen, 2015. Surface Wave Effects in the NEMO Ocean Model: Forced and Coupled Experiments.Journal of Geophysical Research: Oceans 04/2015; DOI:10.1002/2014JC010565

Alari V.,  Staneva, J., Breivik, Ø., Bidlot, J.R., Mogensen, K., Janssen, P, 2016: Surface wave effects on water temperature in the Baltic Sea: simulations with the coupled NEMO-WAM model. Ocean Dynamics, 66, 8, 917-930

Joanna Staneva, Victor Alari, Øyvind Breivik, Jean-Raymond Bidlot, Kristian Mogensen, 2016 : Effects of wave-induced forcing on a circulation model of the North Sea. Ocean Dynamics (2016). doi:10.1007/s10236-016-1009-0

- **The suggested citation has been included at line 596.**

**Reviewer2**

I enjoyed reading this paper because it gives a clear and focused presentation of how the authors developed and validated a three-dimensional wave–current model on unstructured grids. The explanations about how the model is set up and the parameters involved are detailed enough that someone else could feasibly replicate the approach. The validation cases are convincing, showing that the model does a good job of reproducing analytical or observed conditions.

However, one very important validation case is missing, which is the proof of CPU-coherency. The results should be the same for any number of CPUs. In weather centers, there would be a need for binary reproducibility, but for this implementation, I think it is good enough if it is shown that any variation in the results is neglible.

At the same time, I think a little more discussion about computational efficiency and certain assumptions behind the model would help readers understand its practical limits. Overall, though, this is a worthwhile contribution to the field of coastal and ocean modeling, and researchers who deal with complex domains will likely find this work both relevant and informative.

**We thanks the reviewer for the insight he provided. We included an annex to the text where this details are treated.**

There are some small flaws in the manuscript, for example, some references are missing in the text ("! Reference source not found"), and some literature references are used in a way that I think should be improved. For instance, if a reference is given for a finding and it is not clear whether that author was the original source, it would be better to reference (e.g., XXXX et al. XXXX) rather than (XXXX et al. XXXX). Moreover, it is always preferable to cite the original publication of any finding rather than a subsequent replication

**Thank you for pointing out these issues. We have fixed the missing references and revised the citations according to your suggestions. However, if the reviewer has specific recommendations, we are open to incorporating them.**